# *Lactobacillus paracasei* CNCM I 1572: A Promising Candidate for Management of Colonic Diverticular Disease

**DOI:** 10.3390/jcm11071916

**Published:** 2022-03-30

**Authors:** Elisabetta Bretto, Ferdinando D’Amico, Walter Fiore, Antonio Tursi, Silvio Danese

**Affiliations:** 1Gastroenterology and Endoscopy, IRCCS San Raffaele Hospital and Vita-Salute San Raffaele University, 20132 Milan, Italy; bretto.elisabetta@hsr.it (E.B.); damico.ferdinando@hsr.it (F.D.); 2Department of Biomedical Sciences, Humanitas University, 20090 Pieve Emanuele, Italy; 3Sofar S.p.A., Via Firenze 40, 20060 Trezzano Rosa, Italy; walter.fiore@sofarfarm.it; 4Territorial Gastroenterology Service, Azienda Sanitaria Locale Barletta-Andria Trani, 70031 Andria, Italy; antotursi@tiscali.it; 5Department of Medical and Surgical Sciences, Catholic University, 20123 Rome, Italy

**Keywords:** *Lactobacillus paracasei* CNCM I 1572, probiotics, acute diverticulitis, diverticular disease, symptomatic uncomplicated diverticular disease

## Abstract

Diverticular disease (DD) is a common gastrointestinal condition. Patients with DD experience a huge variety of chronic nonspecific symptoms, including abdominal pain, bloating, and altered bowel habits. They are also at risk of complications such as acute diverticulitis, abscess formation, hemorrhage, and perforation. Intestinal dysbiosis and chronic inflammation have recently been recognized as potential key factors contributing to disease progression. Probiotics, due to their ability to modify colonic microbiota balance and to their immunomodulatory effects, could present a promising treatment option for patients with DD. *Lactobacillus paracasei* CNCM I 1572 (*LCDG*) is a probiotic strain with the capacity to rebalance gut microbiota and to decrease intestinal inflammation. This review summarizes the available clinical data on the use of *LCDG* in subjects with colonic DD.

## 1. Introduction

Diverticular disease (DD) is a spectrum of gastrointestinal conditions characterized by the presence of diverticula, defined as small, balloon-like sacs protruding through the layers of the colon [1]. DD is the fifth most important gastrointestinal disease in terms of healthcare costs in Western countries, with the highest rates occurring in the United States and Europe with 60–70% prevalence rates in those older than 60 [1,2]. The presence of one or more diverticula in the colon is called diverticulosis [2,3]. Diverticulosis is a common condition that generally occurs during middle age and remains asymptomatic [2,3]. Its finding is incidental [4]. Approximately 20% of patients with DD develop symptoms, including abdominal pain, bloating, and altered bowel habits, the condition is called symptomatic uncomplicated diverticular disease (SUDD) [1,2]. The major clinical complication, occurring in about 4% of individuals, is acute diverticulitis, that could be uncomplicated or complicated by abscess formation, haemorrhage, and perforation [1,5,6]. Moreover, due to persistent mucosal inflammation, patients with acute diverticulitis may be susceptible to the consecutive development of SUDD [7]. DD appear to share similar pathophysiological mechanisms with irritable bowel syndrome (IBS) [8,9]. In DD, alteration in bacterial microbiota occurs primarily because of faecal material stasis which predisposes to diverticular bacterial overgrowth [10,11,12,13]. This leads to impairment of the mucosal barrier function and up-regulates inflammatory cytokine release with low-grade microscopic inflammation; this inflammation has the potential to progress to microperforation and, ultimately, to acute diverticulitis [10,11,12,13]. In addition, dysbiosis and mucosal inflammation are associated with dysmotility; they alter nerve fibre activation leading to subsequent neuronal and muscular dysfunction, thus favouring the development of abdominal symptoms [14,15].

Given these observations, probiotics may be an appealing treatment option for this condition, due to their ability to modify colonic microbiota balance and to their immunomodulatory effects [16,17,18,19]. Although probiotics have already been proposed for use in inflammatory, infectious, neoplastic, and allergic disorders, the ideal probiotic strain for use in any of these indications has yet to be identified [20,21,22,23,24]. The interpretation of available data on probiotics is further confounded by variability in strain selection, dose, delivery vehicle, and evaluation of viability and efficacy. *Lacticaseibacillus paracasei* (formerly *Lactibacillus paracasei*) CNCM I1572 (*LCDG*; *L. casei DG*^®^; Enterolactis^®^, Sofar S.p.A., Trezzano Rosa, Milan, Italy, deposited at Institute Pasteur of Paris with number I1572) is a probiotic strain with the capacity to rebalance gut microbiota and to decrease intestinal inflammation. This review aims to summarize the available evidence on the use of *L. casei DG*^®^ (*Lactobacillus paracasei* CNCM I 1572—*LCDG*; Enterolactis^®^) in the management of DD.

## 2. Probiotics and Their Mechanisms of Actions

Probiotics are living organisms that are originally found in the intestine. They can also be synthesized in laboratories and therefore be available in commercial products [25,26,27]. Probiotics are defined as live microorganisms that, when administered in adequate amounts, confer a health benefit on the host [27]. The minimum quantity to obtain a temporary colonization of the intestine is generally at least 1 billion live cells per day [28]. Probiotics must also be resistant to gastric acid and bile to survive through the gastrointestinal tract [29,30,31]. Once present in the colon, probiotics must adhere to the colon’s epithelial cells to ensure adequate colonization [32]. Probiotic organisms have the ability to produce antimicrobial substances or antagonize pathogenic bacteria in the gut [33,34]. Finally, commercially manufactured probiotics must be stable for storage before being ingested and must be safe for use in humans’ large amounts [35,36]. The mechanism of action of probiotics is likely to be multifactorial. To simplify, we can resume 3 basic functional principles by which probiotics may confer health benefits: restoration of intestinal microbiota; regulation of immune function; and enhancement of barrier function of gut epithelium (Figure 1) [37,38]. Through competitive inhibition probiotics hinder the ability of pathogenic Gram-negative bacteria to adhere and colonize the intestinal mucosa [39,40,41]. Some probiotic strains secrete proteases that help to degrade toxins [42,43]. Also, by fermenting dietary fibers, probiotics can produce SCFAs (short-chain fatty acids), such as butyrate, propionate, and acetate, with protective properties against intestinal pathogens [44]. Moreover, probiotics have been associated to decreased secretion of inflammatory cytokines such as tumor necrosis factor-α, interferon-γ, and interleukin-1, and increased production of the anti-inflammatory cytokines such as interferon-α and interleukin-10 [45,46,47]. Some species displayed the capacity to stimulate immunoglobulin A secretion in Peyer patches [48]. Finally, probiotics exert a direct effect on the intestinal epithelial barrier function: by binding with toll-like receptors on the apical surface of the epithelial cells [49], they activate protein kinase C within the cell, which results in clenching of the tight junctions between epithelial cells and improving the barrier function of the gut mucosa, thus potentially limiting bacterial translocation [50,51,52].

## 3. *Lactobacillus paracasei* CNCM I 1572

*LCDG* is a Gram-positive bacterial strain isolated from human faeces and normally present in healthy individuals’ intestinal microbiota. It survives the gastrointestinal transit in healthy children and adults when ingested with the probiotic drinkable formulation containing no less than 1 × 10^9^ CFU, demonstrating resistance to digestive juices, hydrolytic enzymes, and bile acids [53,54,55]. *LCDG*’s human origin guarantees a long-lasting intestinal colonization, persisting in the gut of patients up to one week after the end of probiotic consumption [53,54]. *LCDG* produces lactic acid, providing a quick rebalancing action of faecal microbiota [48]. Moreover, *LCDG* does not induce antibiotics resistance, guaranteeing safe human consumption [54]. Consistently with these peculiarities, several in vitro/in vivo experiments demonstrated that the polysaccharides present on the surface of the bacteria, referred either as capsule or as exopolysaccharides (EPSs) can play a role both in the modulation of the intestinal microbial ecosystem and in the stimulation of the host’s immune responses; protecting *LCDG*, these polysaccharides also allow the probiotic strain to reach the intestine alive [48,56,57,58,59,60,61]. Finally, a genomic analysis of *LCDG* revealed that this strain produces a unique rhamnose-rich hetero-exopolysaccharide, named DG-EPS, with the ability to stimulate the production of proinflammatory cytokines by antigen-presenting cells (APCs) responsible for the detection of microorganisms and involved in their clearance through phagocytosis [57]. Acting as mild booster of the innate immunity, *LCDG* may contribute to a more efficient and faster immune response against potential infectious agents [61,62,63].

### 3.1. Lactobacillus paracasei CNCM I 1572 in Human Health

The efficacy of *LCDG* has been investigated in several clinical settings, including healthy humans or patients with urological diseases such as chronic bacterial prostatitis and gastrointestinal disorders such as IBS, small intestinal bacterial overgrowth (SIBO), and DD [48,58,59,60,64,65,66,67,68]. To determine the impact of *LCDG* on the intestinal microbial ecology of healthy patients, Ferrario et al. [48] conducted a randomized, double-blind, placebo-controlled crossover trial on 34 healthy human volunteers’ faecal microbiota. Participants were randomly assigned to 2 parallel groups receiving, once a day for 4 weeks, in addition to their habitual diet, either placebo or a probiotic capsule containing at least 24 billion viable cells of *LCDG*. Each volunteers’ faecal microbiota was evaluated before and after *LCDG*s’ consumption. Despite inter-individual variability in intestinal microbiota, this probiotic strain has been shown to positively modulate microbiota of healthy human, increasing the percentage of bacteria that—according to the literature—can potentially confer a health benefit to the host [69,70,71,72,73,74,75]. In fact, *LCDG* intake induced an increase in the gram-negative phylum Proteobacteria (*p* = 0.006), which is the most abundant phyla in the human gut microbiota, and in the gram-positive Clostridiales genus *Coprococcus* (*p* = 0.009), which play a crucial role in folate biosynthesis and in the colonic fermentation of dietary fiber leading to short chain fatty acids (SCFAs) production [48]. SCFAs (i.e., acetate, butyrate, and propionate) are crucial in preserving gut equilibrium, an increased level of SCFAs leads to the enhancement of barrier function of intestinal epithelium [76]. Butyrate in particular, is linked with a number of beneficial activities on the intestinal mucosa; drastic increase or reduction in its concentration is typical for several pathologies such as IBS and metabolic syndrome [77,78,79,80,81,82,83]. So, modifying the concentration of bacterial groups able to produce SCFAs in the gut, this probiotic strain can “rebalance” SCFAs and butyrate concentrations. After *LCDG* intervention, participants with butyrate > 100 mmol/kg of wet feces had a mean butyrate reduction of 49 ± 21%. In contrast, in participants with initial butyrate concentrations < 25 mmol/kg of wet feces, the probiotic contributed to a 329 ± 255% (*p* > 0.05) increment in butyrate. Finally, a declining trend was observed in genus *Ruminococcus* (*p* = 0.016) known for its role in the etiopathogenesis of IBS [73,84,85]. *LCDG* seems to work in the direction of a potential protective and healthy microbiota, rebalancing gut physiological conditions. Additional studies were then conducted to investigate its application in different colonic diseases. Rosania et al. [65] evaluated the effects of addition of *LCDG* to antibiotics in patients with SIBO during a 6-month follow-up. Twenty patients reporting abdominal compliant without gastrointestinal diseases/alarm symptoms were enrolled. SIBO was diagnosed by the agreement of lactulose and glucose breath tests. Patients received rifaximin 400 mg/day for 7 days/month followed by *LCDG* for 7 days more. All patients recorded a questionnaire for subjective symptom evaluation according to Rome III criteria and Bristol scale for stool characteristics before the study and after 6 months. A significant improvement was obtained in 5 (diffuse abdominal pain *p* < 0.001; pain in the left iliac area *p* < 0.002; meteorism *p* < 0.002; flatulence *p* < 0.001; nausea *p* < 0.01) out of 6 symptoms. The analysis for each single patient also showed an improvement in the number of bowel movements and stool characters in 16 out 20 patients (80%). Besides, in a recent multicenter randomized study, Cremon et al. [58] evaluated the effects of *LCDG* on gut microbiota-related factors in IBS patients’ faecal samples. The participants were randomly assigned to different groups: in one group they had to take *LCDG* two times a day for four weeks, in the other group they had to take the equivalent product without bacteria (placebo), this phase was followed by a washout period of four more weeks before crossing over to the alternate treatment (twice daily for four weeks). After 14 weeks, patients entered a four-week follow-up phase. In all cases, faecal samples were obtained before and after each treatment and follow-up period. The intestinal microbial ecosystem was then characterized. In IBS patients, at baseline, members of the gut microbiota attributed to the genus *Ruminococcus* were increased. Concentration of the proinflammatory cytokine IL-15 was also enhanced, whereas SCFAs levels were decreased. Interestingly, *LCDG* induced a significant trend of reduction in *Ruminococcus* (*p* = 0.042) and in IL-15 (mean change −173.4; *p* = 0.042). In contrast, faecal short chain fatty acids acetate (*p* = 0.021) and butyrate (*p* = 0.047) were increased, confirming once again the potential role of the probiotic strain in such disorders. Finally, to better understand the molecular mechanisms of action of *LCDG*, Compare et al. [59], by using human intestinal biopsy specimens in culture, analyzed the effect of the *Lactobacillus* on ileal and colonic mucosa of 10 post-infectious irritable bowel syndrome (PI-IBS) patients. At baseline, IL-1α, IL-6, IL-8 mRNA levels, and TLR-4 proteins expression were higher while anti-inflammatory IL-10 mRNA levels were lower in PI-IBS patients than in healthy controls. Treatment of colonic biopsies with *LCDG* significantly reduced the levels of all proinflammatory cytokines (Il-1 α *p* < 0.002, IL-6 *p* < 0.0001 and IL-8 *p* < 0.0001) in respect to baseline. In ileal mucosa, *LCDG* treatment was effective in reducing IL-1α and IL-8 mRNA levels (*p* < 0.0002 and *p* < 0.0001, respectively) but did not affect IL-6 levels. In contrast, IL-10 m-RNA levels significantly increased in both ileal and colonic mucosa (*p* < 0.0001 and *p* < 0.0001, respectively). Finally, the increase of TLR-4 protein expression was attenuated by *LCDG* (*p* < 0.0001).

### 3.2. Lactobacillus paracasei CNCM I 1572 and Diverticular Disease

Some new data on the role of *LCDG* in the management of diverticular disease have emerged in the last years. Anti-inflammatory action of *LCDG* on patients with DD was investigated in an in vitro study conducted by Turco et al. [60] Intestinal biopsies were collected during endoscopy in 40 consecutive individuals, divided as follow: 10 patients with diverticulosis, 10 patients with SUDD, 10 patients with SUDD with previous acute diverticulitis (SUDD+AD), and a control group of 10 people without gastrointestinal diseases. Biopsies were then stimulated with the probiotic *LCDG* and/or the pathogen enteroinvasive *Escherichia coli* (EIEC). As previous studies demonstrated an increase in nitric oxide (NO)-mediated responses in patients with DD; the rationale was to evaluate NO release and inducible nitric oxide synthase (iNOS) expression before and after biopsies stimulation [86,87,88,89]. Basal iNOS expression was significantly increased in SUDD and SUDD+AD patients (+2.04- and +2.86-fold increase vs. CTRLs, respectively; *p* < 0.05). Basal NO expression was significantly increased in SUDD+AD (+7.77-fold increase vs. CTRLs; *p* < 0.05) (Figure 2).

In all groups, iNOS expression was significantly increased by EIEC and reduced by *LCDG* (*p* < 0.05 and *p* < 0.05, respectively). In all groups, except for SUDD+AD, EIEC significantly increased NO release and *LCDG* significantly reduced NO release (*p* < 0.05 and *p* < 0.05 respectively). Data confirmed an activation of NO-dependent inflammation related to iNOS expression and NO release that appeared progressively increased from diverticulosis to SUDD with previous diverticulitis. At baseline, a significantly increased release of the anti-inflammatory cytokine IL-10 in patients with SUDD+AD (−11.25+ fold increase vs. controls was also observed; *p* < 0.05), clear evidence of the body’s attempt to control inflammation after acute diverticulitis. Finally, this study demonstrated that colonic mucosa of patients with DD is characterized by a different reactivity towards pathogenic stimuli. *LCDG*s’ role in counteracting the pro-inflammatory effects exerted by EIEC was confirmed, suggesting a beneficial role of this probiotic in DD. Three in vivo studies (Table 1) have then investigated the efficacy of *LCDG* administered in combination with mesalazine (5-ASA), which is commonly used in the treatment of inflammatory bowel disease. The rationale for this approach was to control peri-diverticular inflammation while simultaneously restore local microbiota [90,91,92]. Tursi et al. [66] conducted a multicenter, prospective, randomized controlled study comparing 5-ASA, *LCDG*, and their combination in 90 patients with recurrent SUDD. Subjects periodically self-rated their symptoms using a tool that generated an “overall symptom score”, including constipation, diarrhea, abdominal pain, rectal bleeding, and mucus with the stools, that could range from 0 to 50. After 12 months, 76.7% of subjects (23/30) in each monotherapy arm were symptom-free, compared with 96.7% of those on combination therapy (29/30; one subject lost to follow up). In general, when symptoms occurred, they were rated as mild, but the overall symptom score was lower in the group receiving the combination of 5-ASA plus *LCDG* compared with the other two groups (*p* < 0.001). A related randomized study was reported by Tursi et al. [67] in 75 subjects with additional study arms to assess 5-ASA doses of 0.8 g versus 1.6 g daily; follow up was extended to 24 months. In the present case, 80% and 87% of patients on monotherapy remained symptom-free versus 92% and 94% of those on combination therapy, the differences were not statistically significant. No 5-ASA dose-response was reported. Of note, all subjects who stopped treatment developed symptoms or a diverticular complication. Finally, Tursi et al. [68] conducted another randomized, controlled double-blind, double-dummy trial to evaluate the effectiveness of 5-ASA and/or *LCDG* in maintaining remission in SUDD. The study was conducted on 210 patients with SUDD. Participants were randomly enrolled in four groups: Group M (active 5-ASA 1.6 g/day plus *LCDG* placebo), Group L (*LCDG* 24 billion/day plus 5-ASA placebo), Group LM (active *LCDG* 24 billion/day plus active 5-ASA 1.6 g/day), Group P (*LCDG* placebo plus 5-ASA placebo). Patients received treatment during 12 months for 10 days/month. Recurrence of SUDD was defined as the reoccurrence, for at least 24 consecutive hours, of abdominal pain during follow-up, scored as ≥5 (0: best; 10: worst). Reappearance of SUDD happened in no (0%) patient in group LM, in 7 (13.7%) patients in group M, in 8 (14.5%) patients in group L, and in 23 (46.0%) patients in group P (LM group vs. M group, *p* = 0.015; LM group vs. L group, *p* = 0.011; LM group vs. P group, *p* = 0.000; M group vs. P group, *p* = 0.000; L group vs. P group, *p* = 0.000). Compared with all other groups, in group P there was a significant number of recurrences (*p* = 0.003). Both cyclic 5-ASA and *LCDG* emerged to be better than placebo for maintaining remission in SUDD, particularly when used together (Figure 3). Moreover, both treatments, alone or in combination, were significantly better than placebo in preventing occurrence of acute diverticulitis in SUDD patients. No adverse effects related to the probiotic treatment were observed in the overall studies [66,67,68].

## 4. Discussion

Dysbiosis has been described in patients with symptomatic diverticular disease. Thus, changing gut bacterial composition and switching off inflammatory patterns through probiotics seems an interesting therapeutic strategy, breaking the vicious circle in which dysbiosis and inflammation promote each other. These observations have encouraged to investigate the potential role of *LCDG* as a new therapy for DD [93,94,95,96].

*LCDG* has multiple modes of action (Figure 4), including rebalancing of the intestinal microbiotas’ ecology and regulation of the immune system activity. This probiotic strain has a clear impact on faecal microbiota, modifying specific microbial groups at the phylum and genus levels, inhibiting colonic bacterial overgrowth and metabolism of pathogens, and increasing levels of SCFAs playing an important role in maintaining intestinal homeostasis [48,58]. Moreover, data coming from in vitro and in vivo studies, demonstrated *LCDG*s’ capacity to regulate the immune system activity in IBS and DD, by controlling pro- and anti-inflammatory cytokines levels [58,59,60]. Acting on inflammation, *LCDG* may also act on symptom development in individuals affected by such intestinal diseases [58,65].

In three clinical trials, this probiotic strain seemed to show an apparent trend to significantly obtain remission and reduction of the recurrence of SUDD, especially when used in combination with 5-ASA [66,67,68] (Table 1). However, the poor number of overall studies, the heterogeneous nature, and the relatively poor quality of the available studies on the use of *LCDG* make it difficult to evaluate the cumulative efficacy of this probiotic strain. Treatment protocols about timing, dosage, or combination with other drugs and the follow-up periods in the different studies were very variable. Only 1 study was a double-blinded randomized controlled trial. Furthermore, the type of DD was heterogeneous between the mentioned studies because there is a lack of a globally recognized clinical classification for diagnosing and defining SUDD and its recurrence.

Two main considerations can be extrapolated from this work. First, even if the amount of data present is not sufficient to draw robust conclusions, *LCDG* appears a promising option to promote the health of the individual, equilibrating gut physiological conditions in colonic pathologies such as SIBO, IBS, and DD. Especially in the latter pathology, the efficacy and safety of *LCDG* in improving symptoms of SUDD have already been mentioned in the WGO Global Guidelines of the use of Probiotics and Prebiotics (2017) [97]. It could also be interesting to evaluate its use in inflammatory bowel disease (IBD) and microscopic colitis, two more multifactorial gastrointestinal diseases in which we find the same pathogenetic pattern of chronic inflammation and protracted dysbiosis. Secondly, data emerging from this review suggest that large, randomized placebo-controlled studies are needed to establish efficacy, dose-responses, optimal timing for introduction, and duration of *LCDG* therapy in the spectrum of colonic diverticular disease.

## 5. Conclusions

*Lactibacillus paracasei* is a promising candidate in the management of diverticular disease, ensuring multiple benefic effects on the intestinal homeostasis.

Specific studies to evaluate *LCDG*s’ efficacy in each clinical setting of DD are warranted, including treatment of acute diverticulitis and SUDD, prevention of recurrent diverticulitis, and management of chronic symptoms.

## Figures and Tables

**Figure 1 jcm-11-01916-f001:**
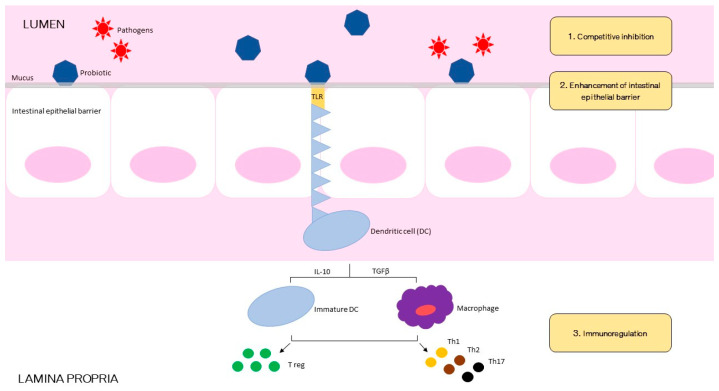
Schematic illustration of postulated mechanisms of probiotic bacterial actions against gastrointestinal pathogenic infection.

**Figure 2 jcm-11-01916-f002:**
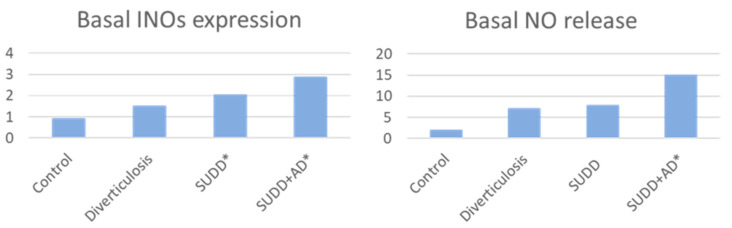
Basal INOs expression and NO release in patients vs. controls * *p* < 0.05 vs. controls INOs = inducible nitric oxide synthase; NO = nitric oxide; SUDD = symptomatic uncomplicated diverticular disease; AD = acute diverticulitis.

**Figure 3 jcm-11-01916-f003:**
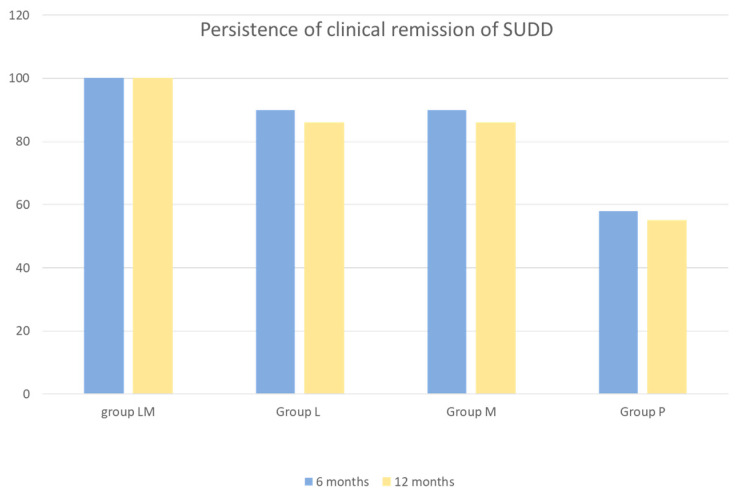
Persistence of clinical remission of SUDD at 6 and 12 months by study group. Absence of recurring abdominal pain (scored ≥ 5 for at least 24 consecutive hours and recorded at any time during the follow up) was defined as clinical remission. SUDD = symptomatic uncomplicated diverticular disease; group LM = active mesalazine + active *Lactobacillus casei*; group L = active *Lactobacillus casei* + mesalazine placebo; group M = active mesalazine + *Lactobacillus casei* placebo; group P = mesalazine placebo + *Lactobacillus casei* placebo.

**Figure 4 jcm-11-01916-f004:**
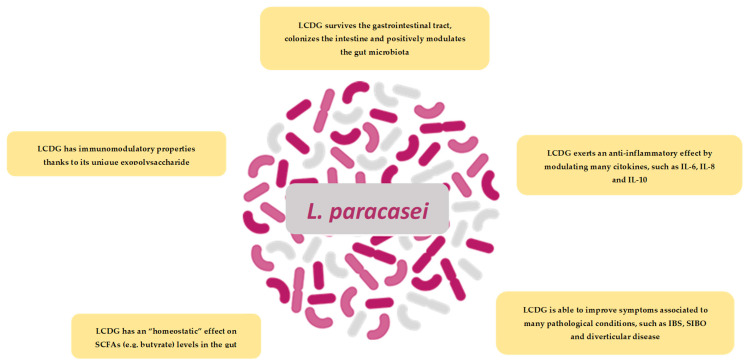
Schematic illustration of the most important finding on *LCDG*.

**Table 1 jcm-11-01916-t001:** Main characteristics of the 3 in vivo selected studies on *LCDG* treatment in DD.

Author Year [Ref.]	# Patients	F	Mean Age (Years)	Type of Diverticular Disease	Study Type	Arms	Single Center Yes/No	Interventions	Follow-Up	Outcome Measure	Efficacy of Interventions
**Tursi et al., 2006 [66]**	85	54	67	Symptomatic uncomplicated DD in remission	Open RT	3	No	G1: 5-ASA 1.6 g/day G2: *LCDG* 8 × 10^9^ CFU 15 days/month G3: *LCDG* 8 × 10^9^ CFU 15 days/month + 5-ASA 1.6 g/day	12 months	Remission of abdominal symptoms	Symptom free at 12 months: G1 76.7% (23/27) G2 76.7% (23/27) G3 96.7% (29/29)
**Tursi et al., 2008 [67]**	75	42	65	Symptomatic uncomplicated DD in remission	Open RT	5	Yes	G1: *LCDG* 16 × 10^9^ CFU 10 days/month + 5-ASA 800 mg/day G2: *LCDG* 16 × 10^9^ CFU 10 days/month + 5-ASA 1600 mg/day G3: 5-ASA 800 mg 10 days/month G4: 5-ASA 1600 mg 10 days/month G5: *LCDG* 16 × 10^9^ CFU 10 days/month	24 months	Remission of abdominal symptoms	Symptom free at 24 months: G1 93.7% (15/16) G2 92.3% (12/13) G3 84% (11/13) G4 80% (8/10) G5 86.9% (20/23)
**Tursi et al., 2013 [68]**	210	101	62	Symptomatic uncomplicated DD in remission	DB placebo-controlled RT	4	No	G1: *LCDG* 24 × 10^9^ CFU 10 days/month + 5-ASA 1600 mg/day G2: *LCDG* placebo + 5-ASA 1600 mg/day for 10 days/month G3: *LCDG* 24 × 10^9^ CFU 10 days/month + 5-ASA placebo G4: *LCDG* placebo + 5-ASA placebo	12 months	Recurrence of abdominal symptoms	Recurrence of SUDD at 12 months: G1 0% (0/54) *p* > 0.01 vs. other arms G2 13.7% (7/51) G3 14.5% (8/55) G4 46.0% (23/50)

Ref. = reference; # = number; F = female; DD = diverticular disease; DB = double blind; RT = randomized trial; G = group; ASA = mesalazine; *LCDG* = Lactobacillus.

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
