# Peer review of "Lactobacillus paracasei CNCM I 1572: A Promising Candidate for Management of Colonic Diverticular Disease"

_jcm, 2022, doi:10.3390/jcm11071916_

Round 1

Reviewer 1 Report

This review demonstrates that Lactobacillus paracasei CNCM I 1572 (LCDG) has a potential to give beneficial effects on patients with colonic diverticular disease, especially when used in the combination with 5-ASA, although, data were based on only 3 clinical trials, containing a double-blind placebo-controlled trial, by same research group. Some revisions are necessary before accept..

Major comments

  1. You have described many positive effects of probiotics in this manuscript. You should properly describe what kind of experimental system (in vitro, in an animal model, in healthy subjects or in patients) the result is, so as not to mislead the reader.
  2. As an action mechanism of probiotics, increased SCFA is considered to be important in the maintaining of intestinal homeostasis, not only functioning as an energy source of colon cells, but also suppressing colonization of pathogenic bacteria, preventing infectious diseases, strengthening the intestinal mucosal barrier, modulating immunity function, and enhancing anti-inflammatory action. The main mechanism of the enhancement of barrier function of intestinal epithelium is generally more widely known to increase SCFA production than to adhere epithelium. So you should describe that the enhancement of barrier function of intestinal epithelium is due to the increased SCFA.
  3. It is an important characteristic that LCDG has extracellular polysaccharides. It is thought to be involved not only in immuno-modulatory action, but also in reaching the intestine alive due to the protection of bacterial cells. You should emphasize this point more.
  4. On the other hand, the ability of lactic acid bacteria with extracellular polysaccharides to adhere to epithelial cells is assumed to be low. The adhesion is considered unnecessary for probiotic requirements, because some lactic acid bacteria have excellent health benefits as probiotics due to their extracellular polysaccharides, even if they do not have the ability to adhere to epithelial cells. Do LCDG that show adhesion to epithelial cells in vitro show adhesion to intestinal epithelial cells in vivo? If there is evidence of adhesion to intestinal epithelial cells in vivo, it should be shown. Without it, you should not describe the theory that adhesion to intestinal epithelial cells is the action mechanism.
  5. Line 138-139, the probiotic contributed to a 329±255% increment in butyrate. Is it statistically significant?
  6. 4, isn't the Immuno-modulative effect and the anti-inflammatory effect just looking at the same effect from different perspectives? It should be unified as one action.
  7. Line 12 in Discussion, what are “different compounds”? It is better to describe the compound name specifically.
  8. The manuscript should be checked by a native speaker as the English description is poor.

Minor comments

  1. Line 201-202, if it is p>0.05, significantly should be revised to no-significantly.
  2. Line 204-206, was the observations at baseline? If it is so, you should add “at baseline”.
  3. Line 20 in Discussion, three in vivo studies ⇒ three clinical trials

Author Response

Response to Reviewer 1 Comments

Comments to the Author

This review demonstrates that Lactobacillus paracasei CNCM I 1572 (LCDG) has a potential to give beneficial effects on patients with colonic diverticular disease, especially when used in the combination with 5-ASA, although, data were based on only 3 clinical trials, containing a double-blind placebo-controlled trial, by same research group. Some revisions are necessary before accept.

Major comments

  1. You have described many positive effects of probiotics in this manuscript. You should properly describe what kind of experimental system (in vitro, in an animal model, in healthy subjects or in patients) the result is, so as not to mislead the reader.

Reply: We thank the reviewer for this suggestion. We described every kind of experimental system as recommended.

  1. As an action mechanism of probiotics, increased SCFA is considered to be important in the maintaining of intestinal homeostasis, not only functioning as an energy source of colon cells, but also suppressing colonization of pathogenic bacteria, preventing infectious diseases, strengthening the intestinal mucosal barrier, modulating immunity function, and enhancing anti-inflammatory action. The main mechanism of the enhancement of barrier function of intestinal epithelium is generally more widely known to increase SCFA production than to adhere epithelium. So you should describe that the enhancement of barrier function of intestinal epithelium is due to the increased SCFA.

Reply: We thank the reviewer for this comment. We added to our manuscript the concept that increased SCFA leads to the enhancement of barrier function of intestinal epithelium.

  1. It is an important characteristic that LCDG has extracellular polysaccharides. It is thought to be involved not only in immuno-modulatory action, but also in reaching the intestine alive due to the protection of bacterial cells. You should emphasize this point more.

Reply: We have appreciated the reviewer’s comment. We revised our manuscript adding this important function of LCDG’s extracellular polysaccharides.  

  1. On the other hand, the ability of lactic acid bacteria with extracellular polysaccharides to adhere to epithelial cells is assumed to be low. The adhesion is considered unnecessary for probiotic requirements, because some lactic acid bacteria have excellent health benefits as probiotics due to their extracellular polysaccharides, even if they do not have the ability to adhere to epithelial cells. Do LCDG that show adhesion to epithelial cells in vitro show adhesion to intestinal epithelial cells in vivo? If there is evidence of adhesion to intestinal epithelial cells in vivo, it should be shown. Without it, you should not describe the theory that adhesion to intestinal epithelial cells is the action mechanism.

Reply: Thank you for pointing this out. Actually there are no evidences that show LCDGs’ adhesion to intestinal epithelial cells in vivo. We revised the manuscript as suggested.

  1. Line 138-139, the probiotic contributed to a 329±255% increment in butyrate. Is it statistically significant?

Reply: We added p>0.05 to our manuscript, which is not statistically significant.

  1. 4, isn't the Immuno-modulative effect and the anti-inflammatory effect just looking at the same effect from different perspectives? It should be unified as one action.

Reply: We appreciate the reviewer’s consideration. We unified both the immune-modulative and the anti-inflammatory effect as suggested.

  1. Line 12 in Discussion, what are “different compounds”? It is better to describe the compound name specifically.

Reply: We have appreciated the reviewer’s comment, we revised the sentence describing specifically the compound name.

  1. The manuscript should be checked by a native speaker as the English description is poor.

Reply: We greatly appreciate the reviewer's advice. The manuscript has been revised by an English native speaker.

Minor comments

  1. Line 201-202, if it is p>0.05, significantly should be revised to no-significantly.

Reply: p>0.05 was a typo, we rectified it to p>0.05 wich is statistically significant.

  1. Line 204-206, was the observations at baseline? If it is so, you should add “at baseline”.

Reply: Yes, the observation was at the baseline, we added it.

  1. Line 20 in Discussion, three in vivo studies ⇒ three clinical trials

Reply: We thank the reviewer for this suggestion. We revised line 20 in Discussion as designated.

Reviewer 2 Report

In the manuscript titled "Lactobacillus paracasei CNCM I 1572: A Promising Candidate for Management of Colonic Diverticular Disease" by Elisabetta Bretto and colleagues, they have reported that probiotics may represent a promising treatment option for patients with diverticulitis disease due to their ability to modify colonic microflora balance and to their immunomodulatory effects. Lactobacillus paracasei CNCM I 1572 (LCDG) is a probiotic strain with the capacity to rebalance gut microbiota and decrease intestinal inflammation. This review summarizes the available clinical data on the use of Lactobacillus paracasei CNCM I 1572 in subjects with colonic diverticulitis disease. I have a few comments regarding the present review manuscript

-The introduction section is a good piece of information, some typos are found in the case of L. casei, the name must be in italics

-I suggest to authors change the word flora for microbiota since 2014 is the preferred word for mention the microbes that reside. 

-Figure 1 needs more resolution, is difficult to see 

-Some typos are found in section 3, the name of L. casei, the CFU exponential number. 

-Figure 2 requires the reference, where this data was presented?

-The same for Figure 4, resolution, and L. paracasei instead L. Paracasei

-Maybe a conclusion section is required in the review article

-In overall the manuscript is a good piece of work, some typos are found, however, the quality is very good, references in the figures for the information that authors have shown are required.

Author Response

Response to Reviewer 2 Comments

Comments to the Author

In the manuscript titled "Lactobacillus paracasei CNCM I 1572: A Promising Candidate for Management of Colonic Diverticular Disease" by Elisabetta Bretto and colleagues, they have reported that probiotics may represent a promising treatment option for patients with diverticulitis disease due to their ability to modify colonic microflora balance and to their immunomodulatory effects. Lactobacillus paracasei CNCM I 1572 (LCDG) is a probiotic strain with the capacity to rebalance gut microbiota and decrease intestinal inflammation. This review summarizes the available clinical data on the use of Lactobacillus paracasei CNCM I 1572 in subjects with colonic diverticulitis disease. I have a few comments regarding the present review manuscript

Comment 1: The introduction section is a good piece of information, some typos are found in the case of L. casei, the name must be in italics

Reply: Thank you for your comment. We apologize for the typos, we have revised our manuscript and made the corrections.

Comment 2: I suggest to authors change the word flora for microbiota since 2014 is the preferred word for mention the microbes that reside. 

Reply: We greatly appreciate the suggestion. We changed the word flora for microbiota.

Comment 3: Figure 1 needs more resolution, is difficult to see 

Reply: Thank you for your comment. We replaced figure 1 and figure 4 whit higher resolution ones, as suggested.

Comment 4: Some typos are found in section 3, the name of L. casei, the CFU exponential number. 

Reply: See the above answer to comment 1.

Comment 5: Figure 2 requires the reference, where this data was presented?

Reply: Figure 2 was created by us extrapolating the data found in the related article.

Comment 6: The same for Figure 4, resolution, and L. paracasei instead L. Paracasei

Reply: See the above answer to comment 3.

Comment 7: Maybe a conclusion section is required in the review article

Reply: We appreciate the advice. We added a conclusion section.

Comment 8: In overall the manuscript is a good piece of work, some typos are found, however, the quality is very good, references in the figures for the information that authors have shown are required.

Reply: Thank you for taking the time to assess our manuscript. We have done the necessary changes as requested.